# Practical Advices for Treating Chronic Pain in the Time of COVID-19: A Narrative Review Focusing on Interventional Techniques

**DOI:** 10.3390/jcm10112303

**Published:** 2021-05-25

**Authors:** Giuliano Lo Bianco, Alfonso Papa, Michael E. Schatman, Andrea Tinnirello, Gaetano Terranova, Matteo Luigi Giuseppe Leoni, Hannah Shapiro, Sebastiano Mercadante

**Affiliations:** 1Department of Biomedical and Biotechnological Sciences, University of Catania, 95124 Catania, Italy; giulianolobianco@gmail.com; 2Anesthesiology and Pain Department, Fondazione Istituto G. Giglio, 90015 Cefalù, Italy; 3Pain Department, AO “Ospedali dei Colli”, Monaldi Hospital, 80131 Naples, Italy; alfonsopapa2@gmail.com; 4Department of Diagnostic Sciences, Tufts University School of Dental Medicine, Boston, MA 02111, USA; Michael.Schatman@tufts.edu; 5Department of Public Health and Community Medicine, Tufts University School of Medicine, Boston, MA 02111, USA; 6Anesthesiology and Pain Management Unit, Azienda Socio Sanitaria Territoriale della Franciacorta, 25032 Chiari, Italy; 7Anaesthesia and Intensive Care Department, Asst Gaetano Pini, 20122 Milano, Italy; dr.gaetano.terranova@gmail.com; 8Unit of Interventional Pain Management, Guglielmo da Saliceto Hospital, 29121 Piacenza, Italy; matteolg.leoni@gmail.com; 9Division of Alcohol, Drugs, and Addiction, McLean Hospital, Harvard Medical School, Boston, MA 02115, USA; hshapiro2@mgh.harvard.edu; 10Pain Relief and Supportive Care, Private Hospital La Maddalena, 90100 Palermo, Italy; terapiadeldolore@lamaddalenanet.it

**Keywords:** COVID-19, chronic pain, pain management, severe acute respiratory syndrome coronavirus 2, telemedicine, analgesics, opioid, spinal cord stimulation, disinfectants

## Abstract

Background: Since the management of chronic pain has become even more challenging secondary to the occurrence of SARS-CoV-2 outbreaks, we developed an exhaustive narrative review of the scientific literature, providing practical advices regarding the management of chronic pain in patients with suspected, presumed, or confirmed SARS-CoV-2 infection. We focused particularly on interventional procedures, where physicians are in closer contact with patients. Methods: Narrative Review of the most relevant articles published between June and December of 2020 that focused on the treatment of chronic pain in COVID-19 patients. Results: Careful triage of patients is mandatory in order to avoid overcrowding of hospital spaces. Telemedicine could represent a promising tool to replace in-person visits and as a screening tool prior to admitting patients to hospitals. Opioid medications can affect the immune response, and therefore, care should be taken prior to initiating new treatments and increasing dosages. Epidural steroids should be avoided or limited to the lowest effective dose. Non urgent interventional procedures such as spinal cord stimulation and intrathecal pumps should be postponed. The use of personal protective equipment and disinfectants represent an important component of the strategy to prevent viral spread to operators and cross-infection between patients due to the SARS-CoV-2 outbreaks.

## 1. Introduction

Severe Acute Respiratory Syndrome Coronavirus 2 (SARS-CoV-2 which can lead to COVID-19 disease) was declared a Public Health Emergency of International Concern on 30 January 2020 and, and as of March 2021, more than 120 million people worldwide have been infected, with more than two million deaths [1].

Pain physicians are treating an ever-increasing number of patients suffering from chronic pain [2]

The most recent data indicate that in Europe, moderate or severe chronic pain affects 22% of the population [3]. In the UK, results of a 2016 community-based population study estimated that 10.4% to 14.3% of the population reported suffering from moderate to severely disabling chronic pain and more than half of the elderly population claimed that chronic pain is the factor which most affects their quality of life [4,5]. Hence, satisfactory and tailored chronic pain management is a priority both morally and ethically, helping maintain patients’ quality of life and protecting against subsequent psychological and physical complications [6,7,8]. During the SARS-CoV-2 pandemic, due to the reallocation of public health resources and the emergence of a series of complex needs and urgent requirements [9,10,11], the need for comprehensive chronic pain management has become even more challenging.

In countries in which the authors work (Italy, UK, USA), hospital-based chronic pain management activities were almost completely suspended from March until June 2020 and have then gradually been resumed. Physicians face the need to deliver high-quality treatments while ensuring safety for patients and health care workers while preventing infection spreading and contamination.

The high transmission rate of SARS-CoV-2 implies a rigorous platform of safety surveillance and meticulous organization in order to avoid further spreading of the disease and hospital outbreaks of infection while allowing for care of chronic pain patients.

Accordingly, the aim of this manuscript is to provide practical advice on the management of chronic pain in patients with suspected, presumed, or confirmed diagnoses of SARS-CoV-2 infection. Additionally, recommendations for hygienic maintenance of the clinic and its equipment during this challenging time are provided. Particular points of focus include: (A) interventional pain specific techniques under fluoroscopy/ultrasound guidance; (B) opioid use among COVID-19 patients; (C) telemedicine provision to chronic pain sufferers; (D) preventive measures to adopt for SARS-CoV-2 infected patients, both in acute and chronic pain settings; and (E) the psychological impact of COVID-19 among both patients and physicians involved in pain management.

## 2. Materials and Methods

The first author (GLB) identified and invited pain physicians and psychologists to join an expert panel to develop practical advice. All panel members were engaged in caring for patients with chronic pain and had experience and training in clinical research in secondary and tertiary care settings. Panel members were interviewed and asked to summarize the most relevant articles published between June and September of 2020 that focused on the treatment of chronic pain in COVID-19 patients. The literature search was conducted using the PubMed, MEDLINE/OVID, and SCOPUS databases. Each author selected relevant articles in their area of major expertise (as detailed in authors’ contribution) Based on the present pathophysiological understanding of COVID-19 and potential practice implications according to the complex management of chronic pain, the panel developed its practical advice in this comprehensive narrative review with the purpose of summarizing the most relevant point of focus regarding chronic pain management during SARS-CoV-2 outbreaks.

## 3. Results

### 3.1. Infection Prevention and Control

SARS-CoV-2 [12], a small lipid-based enveloped virus that belongs to the coronavirus family.

Coronaviruses such as SARS and Middle East Respiratory Syndrome (MERS) can survive on dry inanimate surfaces such as metal, glass and plastic (and ultrasound systems) for between 48 and 96 h [13,14]. SARS coronavirus, MERS coronavirus or endemic human coronaviruses have been shown to persist on fomites for up to 9 days, and therefore, this is an important consideration for ultrasound and other equipment used in all clinical settings [14]. The survival time of the virus depends on material, temperature, humidity and viral concentration. Temperature plays a fundamental role in decreasing viral survival time, as raising temperatures above 56 °C significantly decreases viral concentrations in 10 min, and above 70 °C, viral material is undetectable in just 5 min [14]. However, these data refer to viral RNA detection which is not necessarily strongly correlated with actual infectiousness; therefore, the mere detection of viral genomic material does not necessarily imply the risk of transmitting the disease. Results of a 2003 study on coronavirus strains from a SARS outbreak suggest that the risk of infection by contact with a droplet contaminated paper is small [15]. Hand washing after touching potential materials is therefore considered an effective safeguard against SARS-CoV-2 transmission [15].

There is a paucity of data regarding SARS-CoV-2 inactivation by disinfectants [16]. The United States Centers for Disease Control and Prevention (CDC) recommends social distancing, community mask-wearing, and practicing appropriate hand hygiene as the most effective strategy at this juncture to reduce COVID-19 infectivity rates [17]. The CDC advocates for the use of alcohol-based hand sanitizers (ABHS) while, in their last update on 17 May 2020, they providing warnings against products containing benzalkonium chloride (BAC), stating: “BAC, along with both ethanol and isopropanol, is deemed eligible by the FDA for use in the formulation of healthcare personnel hand rubs. However, available evidence indicates BAC has less reliable activity against coronavirus than either of the alcohols” [17].

Protecting equipment such as ultrasound machines and intrathecal pump (ITP) programmers against contamination should be undertaken through appropriate use of covers. In addition, it is important to ensure that any medications (e.g., ITP refills) and equipment should be transported fully covered in plastic bags. Furthermore, it is essential that these bags and their contents are handled with sterile gloves in a sterile area. Since ultrasound gel is easily contaminable, the use of single-use gel packets is preferred. If these are not available, gel bottles can be used, ensuring that they are not refilled and that their lids are closed throughout exams [13,14].

### 3.2. Screening

We recommend performing a telephone screening the day prior to non-urgent procedures (as illustrated in Table 1) to assess possible risk factors for SARS-CoV-2 infection (i.e., respiratory symptoms, fever, cough, anosmia, dysgeusia in the last three weeks or possible contacts with infected subjects) and postponing treatment of patients with CoV-2-risk factors until viral testing results are obtained, confirming negativity [18]. A standardized triage regarding symptoms and potential contacts with infected subjects should be completed during the telephone screening and signed by the patient during his/her consultation [18]. Patients with known positivity or strongly suspected of active SARS-CoV-2 infection should be treated only for urgent, non-deferrable procedures (as specified in Table 1) and scheduled at the end of the day in order to minimize risk of contamination to other patients.

### 3.3. General Precautions and Personal Protective Equipment (PPE) Requirements

As a suspected or confirmed COVID-19 patient enters the hospital or clinic, prevention of spread of infection must be a priority. The use of Personal Protective Equipment (PPE) is crucial for healthcare personnel (HCP) and should be done in conjunction with general hygiene rules, with particular emphasis on hand sanitizing [19,20,21].

Conscientious use of PPE reduces exposure to body fluids and infectious agents [22].

Preventive measures should include:All patients must wear a surgical mask for the duration of the intervention; [23]Only the patient and treatment team should be allowed into the examination room;A maximum of 1 accompanying person can be admitted in special circumstances (e.g., elderly or patients with impaired mobility);Limiting the number of health care personnel in the examination room;Students and trainees should not be permitted to enter.

Interventional pain procedures can be categorized along 3 main dimensions: short or prolonged neuraxial entry, percutaneous or incisional/surgical procedures, and aerosol-generating or non-aerosol-generating procedures. Standard precautions using sterile techniques are appropriate for interventional pain procedures and surgeries, provided that adequate testing can be performed [13,24]. When aerosol-generating procedures are performed, airborne precautions are also suggested [25,26,27]. In patients with SARS-CoV-2, all of these precautions should remain in place; however, the addition of an eye shield or face shield is also essential in order to observe droplet precautions. Furthermore, during the present crisis, it is recommended that the remaining operating room (OR) staff enter the OR approximately 15–30 min after intubation, depending on the available air-exchange rate.

### 3.4. Care and Cleaning of Rooms and Equipment

As commonly recommended, during the outbreaks secondary to SARS-CoV-2, examination rooms should be free from all non-essential objects. Additional precautions such as cleaning of all furniture, ultrasound probes, keyboards, touch-screens and gel bottles with a low-level disinfectant (LLD) after each examination should be implemented. When covering ultrasound equipment, using a single-use probe cover is recommended.

### 3.5. Interventional Fluoroscopy Procedures

The global COVID-19 pandemic has significantly modified interventional pain management of chronic pain patients. Indeed, during the early phase of the pandemic, elective surgery to relieve pain was temporarily postponed, and the use of telemedicine was suggested. The American and European Societies of Regional Anesthesia and Pain Medicine (ASRA, ESRA) had recommended performing only urgent procedures such as ITP refills and device malfunction or infection management (Table 1 and Table 2) [28].

Adherence to this guidance eased as infection rates decreased, but given the recent dramatic surge in cases worldwide, this recommendation once again becomes quite salient. The evaluation of semi-urgent procedures (as specified in Table 1) is made by a pain physician, with conditions such as intractable cancer pain, acute or subacute herpes zoster or intractable postherpetic neuralgia, acute disc herniation with radiculopathy, intractable trigeminal neuralgia, complex regional pain syndrome and acute cluster headaches falling into this category according to current guidelines [29].

With regard to alternative strategies, such as various forms of neuromodulation [30,31,32] Deer and colleagues [29] recently reported that Australia has allowed radiofrequency ablations without prior diagnostic blocks and spinal cord stimulator implantation without the need for an external trial. Moreover, as recently noted [33,34,35], choosing a non-rechargeable implanted pulse generator (IPG) directly implanted without a trial phase resulted in effective spinal cord stimulation for the management of neuropathic pain in a pilot study. The use of telemedicine for most neurostimulation device troubleshooting was highly recommended, when possible, to solve device malfunction, modify stimulation patterns and manage technical problems or hardware malfunctions.

According to the approach developed by Thomson and colleagues [34], physicians can consider a pre-implant score to select clinical conditions or targeted procedures [35] with high probability of a successful spinal cord stimulation (SCS) trial. In cases of loss of neurostimulator function due to lead migration, lead fracture and IPG malfunction, the International Neuromodulation Society (INS) suggested avoiding surgically revising neurostimulator implants until planned elective surgeries are re-initiated [36,37]. The same authors recommend against the implant of any new ITP system with the exception of judiciously selected cancer pain cases in which the benefit is considered to outweigh the risk of acquiring a COVID-19 pneumonia by using adequate personal protective equipment (PPE). ITP refills should be ensured in order to prevent drug withdrawal as well as unnecessary severe pain exacerbations. Programmable ITP should be surgically replaced every 6–10 years depending on battery consumption, and careful planning of elective ITP replacement has been suggested, reserving surgery for devices whose battery exhaustion is imminent. [36]. Moreover, since ITP refills are procedures that require the operator to come within a distance of less than 1 m from the patient, the PPE used should be commensurate with local guidelines. In selected cases or in high-risk patients in which ITP infusions are solely of opioids, oral equivalents can be substituted. The possible onset of withdrawal symptoms should be considered and adequately managed, since opioid equivalence is not an exact estimation [38]. To the contrary, the oral substitution of baclofen or clonidine is not suggested due to potential life-threatening withdrawal effects.

Corticosteroids are commonly used in interventional pain management for their anti-inflammatory properties. The frequently used routes of administration are via epidurals, joint blocks, peripheral nerves and soft tissue injections. The major anti-inflammatory effect is the result of phospholipase inhibition with a subsequent reduction of cytokine expression, a reduction in the chemotactic or chemoattractant properties of lymphocytes and membrane stabilization [39]. Although the rationale for epidural steroid injections relate to their anti-inflammatory properties, it has been suggested that their perceived effect is also based on blocking conduction in nociceptive nerve fibers [40]. Regarding viral infections, joint corticosteroid injections were associated with a significant increase in the risk of infection, even among flu-vaccinated patients [41]. Since Rabinovitch and colleagues [42] reported a strong correlation between epidural volume and pain relief irrespective of steroid dose for up to one year, it can be argued that epidural steroid injections (ESI) are not the only important component of the sound treatment of epidural inflammation. Moreover, as recently demonstrated by Bise and colleagues, platelet-rich plasma is not inferior to ESI in terms of pain reduction and Oswestry Disability Index (ODI) improvement for patients with persistent radicular pain (>6 weeks) [43]. In early March of 2020, the Royal College of Anaesthetists [44] recommended against the use of systemic corticosteroids in patients suffering from or at high risk of COVID-19, since the immunological impacts of ESI and the effects of steroids use in interventional pain management for patients with this new virus are still unknown. To the contrary, in early June of 2020, the preliminary results of the RECOVERY trial demonstrated a 3% mortality reduction in patients on mechanical ventilation and treated with dexamethasone 6mg once daily for up to ten days compared to usual care [45]. Consequently, the Spine Intervention Society (SIS) alerted interventional pain physicians regarding a possible dexamethasone shortage and prioritizing procedures, by weighing risks and benefits for each individual patient (SIS Guidance on Interventional Pain Procedures During the COVID-19 Global Emergency) [46].

### 3.6. Interventional Ultrasound Procedures

Ultrasound has become a mainstay of pain management, both for patients’ evaluations and for the performance of interventional techniques with precision and safety. Regarding the risk of transmitting infectious diseases, ultrasound-guided procedures range from minimally invasive to critically invasive. Pain management procedures under ultrasound guidance can be considered minimally critical and semi-critically invasive procedures; they may involve micro-trauma to the skin and mucosal membranes, particularly when performing intra-articular injections or deep injections into spinal structures such as facet joints or the epidural space.

Ultrasound probes are sensitive devices and may be damaged with the use of certain chemical compounds, in particular cleaning that is performed with the use of alcohol-based products [47]. SARS-CoV-2 is considered the least resistant to inactivation by common disinfectants used in LLD based on quaternary ammonium compounds [48]. The structure of these viruses includes a lipid envelope, which is easily disrupted with 1 min by most disinfectants such as 62–71% ethanol, 0.5% hydrogen peroxide or 0.1% sodium hypochlorite. Other biocidal agents such as 0.05–0.2% benzalkonium chloride or 0.02% chlorhexidine digluconate are less effective [49]. HLD (high-level disinfectant) is based on glutaraldehyde-based formulations, highly concentrated hydrogen peroxide (7.35%) or 0.23% peracetic acid. These substances can effectively remove all bacteria, fungi or viruses from surfaces. For non-critical devices and for general room and materials disinfection, as the risk of infection transmission is low, ultrasound transducers can be cleaned and disinfected using an LLD or intermediate-level disinfectant; both will denature most bacteria, some fungi and some viruses, such as SARS-CoV-2, influenza A and human immunodeficiency virus [50]. No specific guidelines or recommendations have been published regarding the use of ultrasound for pain management interventions in patients with suspected or confirmed SARS-CoV-2 infection. Thoroughly cleaning environmental surfaces with water and detergent and applying commonly used hospital-level disinfectants (such as 0.1% sodium hypochlorite) are considered effective and sufficient procedures. The World Federation for Ultrasound in Medicine and Biology issued a position statement regarding the safe performance of an ultrasound exam and proper care of the equipment during the pandemic period [51]. In order to maintain the highest degree of safety for both patients and physicians, the following measures should be considered:Screening of patients with possible SARS-CoV-2 infection;Protection of the patient and healthcare personnel;Proper caring and cleaning of examination rooms and equipment.

Moreover, the number of patients moving through hospital facilities should be limited, and the possibility of cross-infectivity among clinical staff members should be minimized. For example, in Italy, hospitals have been formally mandated to develop a strategic plan to organize appointments, surgical planning and clinical staff surveillance. Thus, every patient scheduled for elective day-hospital procedures or surgeries must undergo a test for SARS-CoV-2 detection in the 72 h prior to being admitted to the hospital and, if positive, the procedure will be postponed and the patient sent home to quarantine. Body temperature is checked for every individual (staff and patients) entering the facility and anyone positive for a fever > 37.5 °C or respiratory symptoms (cough, rhinorrhea, anosmia, shortness of breath) is denied access and undergoes testing for SARS-CoV-2. The duration of appointment slots has also been expanded in order to allow sufficient time for disinfection and cleaning following each procedure.

Interventional pain procedures reduced opioid consumption [52], which is a cause of immune-suppression, which predisposes individuals to develop COVID-19 disease. Furthermore, interventional procedures can improve the quality of analgesia, provided that they are provided judiciously through evaluation on a case-by-case basis, ideally involving interdisciplinary team discussion [53,54].

### 3.7. Opioids

The association between chronic pain and the immune system, and pain’s ability to induce immunosuppression, has long been recognized [55]. Opioids are known to cause serious adverse events in some patients, including modification of the endocrine system and immunosuppression [56,57,58]. In fact, opioids can interfere with the innate and acquired immune response by acting on the hypothalamic-pituitary-adrenal axis and the autonomic nervous system; therefore prolonged therapy and higher dosages may intensify endocrinologic disorders [59,60].

However, it is essential to note that various opioids differ in their effects on the immune system, with morphine and fentanyl having the greatest immunosuppressive action and buprenorphine the weakest [61,62,63,64,65,66,67]. Research suggests a potential increase in the incidence and severity of lung infection in patients on chronic long-acting, high-dosage opioids [68,69,70,71], although these studies were not focused on viral infections. Even if SARS-CoV-2 has a profound impact on the immune system, there are no clinical or experimental data regarding increased severity of the disease associated with concomitant opioid utilization. Therefore, no clear recommendations can be made regarding possible suspension or modification of current treatment in patients with chronic opioid exposure [72]. Hence, pain physicians should consider making changes to opioid therapy regimens only subsequent to in-person evaluation of current treatment. This should include obtaining a thorough history and a physical examination. However, due to the COVID-19 health emergency and related distancing measures, physicians may not be able to follow some of the recommended practices.

Taking these factors into consideration, therapy may include administration of short-term opioids to patients experiencing acute pain episodes or severe chronic pain aggravation. This is only advised assuming careful risk stratification and appropriate screening for signs likely to signify potential risks for aberrancy. Furthermore, consultation with the prescription drug monitoring program and an exit strategy with which patients are in agreement should be the standard of care. In patients requiring opioid therapy for a period of more than 1 or 2 weeks, an in-person examination is recommended within 4 weeks to allow physicians to assess the severity of the pathology through physical examination, if possible [72,73].

For patients already receiving high-dosage, long-term opioid therapy, increased opioid treatment for a limited period may be suggested, provided that appropriate risk mitigation practices are utilized. However, we recommend an in-person visit, other than in situations in which doing so is not feasible, within 8 weeks to help identify an advance of the disease process requiring treatment or signs of opioid tolerance or opioid-induced hyperalgesia [73].

Interactions between opioids and antiviral medications should be considered in patients with SARS-CoV-2 infections. Despite conflicting data, Lopinavir/Ritonavir and Remdesivir have been used as a standard aspect of infection treatment for patients admitted with COVID-19 disease. These medications have a profound impact on metabolism of opioids, with Ritonavir inhibiting CYP3A4, which is a key aspect of most opioids’ metabolic pathways. Further, oxycodone plasmatic levels are greatly increased with concomitant lopinavir use [74], thereby increasing the risk of respiratory depression and overdose. On the other hand, methadone plasmatic levels have been determined to dramatically decrease with concomitant use of Lopinavir/Ritonavir (possibly due to an induction of methadone metabolic clearance, involving either or both CP450 3A and CYP450 2D6). Induction of other enzymes, such as intestinal glycoprotein P-450, could also contribute to decreases in drug levels, thus increasing the likelihood of withdrawal syndromes [75]. Remdesivir seems to have less pronounced interactions with other drugs, making it a safer choice in patients taking multiple medications. Morphine, buprenorphine, or tapentadol (drugs whose metabolism is not dependent on CYP450 enzymatic activity [76] could potentially be safer in patients on antiviral therapy. In patients for whom opioid rotation is not possible, careful dosage adjustment is strongly recommended [77].

### 3.8. Telemedicine

Telemedicine refers to the electronic exchange of medical information through a variety of platforms including telephone consultation, video conferencing and short message services for the delivery of health care services remotely. It is used throughout most of the western nations and seeks in general terms to be analogous to traditional care [78]. Research has demonstrated that patients using telemedicine are highly satisfied with telehealth services, appreciating the comfort and convenience offered [79,80]. These data supporting telemedicine’s efficacy include those for patients suffering from chronic diseases or requiring post-procedural follow-up. The time savings and reductions in cost are undoubtedly of considerable interest to health organizations that have increased their reliance on telehealth services [79,80].

Following the appearance of SARS-CoV-2, the role of telemedicine [81] has become of fundamental importance, not only in an attempt to mitigate the spread of the disease, but also as a means of reducing the consumption of personal protective equipment and preserving it for physicians on the front lines. Due to the fact that clinicians are increasingly required to use remote strategies to clinically assess patients, in-person visits have become limited only to those of extreme urgency. One such strategy is the use of patient-reported outcome measures (defined as any report of the status of a patient’s health condition that comes directly from the patient) which can be carried out remotely through the use of mobile phones with cameras, and also as a means of sharing images of paper assessments. Furthermore, the electronic administration of measures is already an integral part of many electronic health record systems used in treating patients with pain. For example, the CHOIR system in the United States [82] or PAIN OUT in Europe [83] are web-based systems which have been specifically adapted for chronic pain sufferers.

Telemedicine is also useful for streamlining a series of procedures, such as patient procedural education, preauthorization prior to performing procedures, pre/post-procedural consultation, and intermittent remote outcome monitoring. Even aspects of the physical examination itself can be carried out via video conferences, e.g., when judging appearance and movement, or when conducting self-examination under guidance. An element of considerable interest is the development of telemedicine in determining which patients are emergent, rather than urgent or elective [84], thereby allowing clinicians to prioritize which procedures to perform. Telemedicine communication, by using real-time interactive audio-visual communication systems, could also be useful for monitoring patients for opioid withdrawal; checking for an elevated heart or pulse rate, for example, which are classic signs of opioid withdrawal. Although promising, there are concerns related to the empirical evidence supporting remote monitoring. For example, relatively few studies have assessed telehealth for potential harm, and dropout rates related to telehealth can be high due to patients’ perceptions that telehealth is inferior to in vivo treatment because it is less “personal” [85] and is subjected to digital discrimination [86,87].

As aforementioned, a practical example in which use of remote monitoring or more generally, of telemedicine, is suggested, are the programming and control of spinal cord stimulators remotely or opioid addiction control or dosage adjustment.

### 3.9. Psychological Considerations

During the COVID-19 pandemic, a high incidence of psychological distress and symptoms have also been observed, including mental health disorders, posttraumatic stress disorder, psychosomatic disorders, and substance abuse [88,89,90].

Many suffer from affective disorders (particularly depression), while others suffer from substance abuse, personality disorders, and various somatoform disorders such as conversion, hypochondriasis, and somatization disorder (not to be confused with “somatization” as a normal process). In some patients, certain of these varied disorders may be secondary to chronic pain, but in others they predate the onset of pain or reflect alternative expressions of the same underlying psychobiological disorder [91].

Furthermore, individual and societal disruption associated with COVID-19 are likely to increase their likelihood of emergence [89,92]. Moreover, while the prevalence of Borderline Personality Disorder is inordinately high among chronic pain patients, the severity of the disorder itself is likely to be increased during the COVID-19 outbreak [93]. A failure to address these issues has the potential to adversely impact pain-related treatment outcomes [94,95]. This has created the need to deliver immediate mental health screening and treatment interventions to large populations, with concerns regarding the supply of adequately trained mental health clinicians arising. [95] Fortunately, telehealth lends itself well to psychotherapeutic approaches [96], and reports of their success during the ongoing COVID-19 crisis are already emerging [97,98].

As recommended by the WHO, a practical example of psychosocial support during SARS-CoV-2 outbreak would be that provided to older adults. Although always a vulnerable population, those living in isolation and those with cognitive decline/dementia, are prone to becoming more anxious, angry, stressed, agitated and withdrawn during the outbreak or while in quarantine [99] older adults.

## 4. Discussion

Due to the COVID-19 pandemic, there exists an increased risk of chronic pain patients failing to receive critical treatment. Chronic pain patients may also be at increased risk of COVID-19 disease due to multiple factors, such as chronic opioid therapy potentially making them more susceptible to the COVID-19 infection due to immunosuppression [54,55,56,57,58,59]. As outlined, added precautions relating to appropriate social distancing and more conscientious sanitization processes in hospitals and clinics need to become a greater focus for those treating patients with pain. Triage of pain patients, while always important, becomes even more imperative due to the need to distinguish between those who may be adequately treated via telemedicine and those requiring in-clinic consultations.

Based on the limited extant literature in conjunction with our clinical experiences, we suggest that interventional pain management can be reinitiated, albeit cautiously, to more effectively treat chronic pain patient population. As steroids are associated with immunosuppression, as well, throughout the remainder of the COVID-19 pandemic, epidural steroid injections should be performed judiciously and with the lowest possible effective dose [100]. SCS and ITP difficulties or technical problems should, when possible, initially be addressed remotely, with in-person visits only in cases of infection or other emergencies. A key element for the future should be even more conscientious planning of pain management with appropriate patient selection. Without exception, efforts should be geared toward enhancing safety conditions in order to protect patients’, physicians’, and support staff’s health and well-being. Given that the duration of the COVID-19 pandemic is uncertain, pain clinicians can adopt “new best practices” that may allow them to treat patients with pain now, as well as more safely and effectively in the future. This review has several limitations, while we analyzed the relevant publications and recommendations that we reviewed between June and December of 2020, we are aware that this review may not be completely exhaustive given that the knowledge on this topic is evolving. We mainly focus on the organization of the clinical practice and did not cover specific clinical topics. The authors work in different countries and regions where regulations, hospital organization, security and screening protocols are different, therefore some recommendations could not be applied everywhere.

However, we hope that this review serves as source of guidance to chronic pain clinicians in the future, and believe that it will remain relevant, irrespective of the course of the pandemic.

## Figures and Tables

**Table 1 jcm-10-02303-t001:** Interventional Pain Procedures.

Urgent Examples	Emergent Examples
Neurolytic procedures for refractory cancer pain	Implanted patients with wound complications
ESI for acute disk herniation	Epidural blood patch for refractory spinal headache
Replacement of neurostimulation devices if therapy cessation leads to abrupt decompensation	Migration of SCS or DRG leads with neurological deficits
Sympathetic blocks for early CRPS in refractory patients	Intrathecal pump refill of malfunction
	Epidural or paravertebral catheter for rib fractures

ESI, epidural steroid injections; CRPS, complex regional pain syndrome; SCS, spinal cord stimulation; DRG, dorsal root ganglion.

**Table 2 jcm-10-02303-t002:** Interventional Pain Procedures.

Elective Procedures	Postpone
Urgent procedures	decide on case-by-case basis
Emergent procedures	proceed with caution, and, if possible, at the end of surgery list

## Data Availability

Literature is available at the following websites: https://pubmed.ncbi.nlm.nih.gov (accessed on 17 September 2020); https://medlineplus.gov (accessed on 17 September 2020); https://www.scopus.com/home.uri (accessed on 17 September 2020).

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
