# Peer review of "Practical Advices for Treating Chronic Pain in the Time of COVID-19: A Narrative Review Focusing on Interventional Techniques"

_jcm, 2021, doi:10.3390/jcm10112303_

Round 1

Reviewer 1 Report

This manuscript provides an up-to-date review on chronic pain management during COVID pandemic for patients with suspected, presumed or confirmed covid 19 infection. This is a timely review as currently there is no guideline for clinicians on how best triage chronic pain patients during a pandemic. Use of telemedicine is a promising tool to replace face to face visits. Principles of use of opioid and epidural steroid were also reviewed. The authors also presented strategy on PPE and disinfection process. This manuscript is well organized and has meaningful clinical implication. However, I have the following concerns

  1. I suggest that authors break down the chronic pain conditions by sites: headache, back pain, cancer related pain, fibromyalgia etc. Headache is the 2nd most disabling disorders worldwide. Headache management is not the same as in general chronic pain management. Headache is highly prevalent, especially during COVID.
  2. Psychological consideration (3.9) The importance of psychological comorbidities in chronic pain should be emphasized, new strategies are being developed by psychologist and psychiatrist, more review on this would be necessary. If the article is mostly focused on the interventional procedures for chronic pain, then the title and abstracts should reflect that. 

Author Response

  1. I suggest that authors break down the chronic pain conditions by sites: headache, back pain, cancer related pain, fibromyalgia etc. Headache is the 2nd most disabling disorders worldwide. Headache management is not the same as in general chronic pain management. Headache is highly prevalent, especially during COVID.                                                                We decided to divide the article not based on clinical diagnosis or anatomical district to reflect our aim to provide  to be used in daily clinical practice and hospital organization
  2. Psychological consideration (3.9) The importance of psychological comorbidities in chronic pain should be emphasized, new strategies are being developed by psychologist and psychiatrist, more review on this would be necessary. If the article is mostly focused on the interventional procedures for chronic pain, then the title and abstracts should reflect that.  Since we mainly focused on interventional procedures we changed the abstract and the title accordingly

Reviewer 2 Report

This narrative review aims to provide practical advices regarding the management of chronic pain and additionally recommendations for hygienic maintenance of the clinic and its equipment during COVID 19. Invited physicians and psychologists joined an expert panel and recommended relevant literature. The results showed the following: Triage of patients is mandatory, telemedicine could be promising, opioids should be minimized, epidural steroids should be avoided, major interventional procedures should be postponed, personal protective equipment and disinfectants is an important component to prevent the spread of virus

In general: Thank you for a very interesting study dealing with an important topic.

Why did you chose to mix chronic pain and hygiene? Please explain your choice more clearly. I find it hard to draw a line between the two themes when reading the aims. Is the hygienic recommendations for chronic pain patients special compared to a non-chronic pain population? Please explain.

Abstract: Contains more than 200 words. Line pp 1 L. 28 in the abstract seems to miss the word “management” after …chronic pain.

Materials and methods: What defined “most relevant” articles? How many participants did you have in your expert panel? Males/females? How many articles could each expert recommend? Did all of the panel experts cover both the primary and the secondary outcomes when they recommend articles?

Results: Please begin by answering your primary outcome. Thereafter, the secondary outcomes.

In the paragraph 3.2 pp 3 line 133-143 there are no references to underpin your recommendations, please add.

I think paragraph 3.3 and 3.4 should be erased or shortened considerably, since it is common knowledge and not in need of an expert panel’s opinion.

Discussion: The discussion section is not quite a discussion but more like a conclusion, maybe just call it a conclusion. I think it would sharpen the article if a box were added with your future recommendations/directions in, described in short with bullet points.

Pp10 line 477-479. Again, what were relevant literature? Was it the experts alone or in combination with you, the authors, who selected the literature? Who decided if the literature was useful? Please explain in the method section.

Please describe the limitation of the study more clearly.

Author Response

Why did you chose to mix chronic pain and hygiene? Please explain your choice more clearly. I find it hard to draw a line between the two themes when reading the aims. Is the hygienic recommendations for chronic pain patients special compared to a non-chronic pain population? Please explain.

Our considerations on hygiene could be applied also to other patients accessing an hospital service but chronic pain management poses specific issues since it involves the use of either potentially immunosuppressive medications (opioids and steroids) and interventional procedures in particularly frail patients. 

Abstract: Contains more than 200 words. Line pp 1 L. 28 in the abstract seems to miss the word “management” after …chronic pain.

We revised and shortened the abstract

Materials and methods: What defined “most relevant” articles? How many participants did you have in your expert panel? Males/females? How many articles could each expert recommend? Did all of the panel experts cover both the primary and the secondary outcomes when they recommend articles?

Most relevant were defined as articles representing scientific societies guidelines, consensus papers,  papers with specific topics on pain management and SARS CoV-2 infection.

The expert panel are the authors of this paper, seven males and one female.

Author contribution has been expanded and we highlighted the specific area of expertise of each author.

Results: Please begin by answering your primary outcome. Thereafter, the secondary outcomes.

In the paragraph 3.2 pp 3 line 133-143 there are no references to underpin your recommendations, please add.

We added a reference

I think paragraph 3.3 and 3.4 should be erased or shortened considerably, since it is common knowledge and not in need of an expert panel’s opinion.

We shortened these paragraphs leaving only essential informations.

Discussion: The discussion section is not quite a discussion but more like a conclusion, maybe just call it a conclusion. I think it would sharpen the article if a box were added with your future recommendations/directions in, described in short with bullet points.

The article format required a discussion section, this can be considered just a summary of all recommendations.

Pp10 line 477-479. Again, what were relevant literature? Was it the experts alone or in combination with you, the authors, who selected the literature? Who decided if the literature was useful? Please explain in the method section.

All the authors selected the most relevant articles in their area of expertise, we provided a more detailed description in the method section and authors contribution.

Please describe the limitation of the study more clearly.

We expanded the limitation section at the end of the paper

Round 2

Reviewer 1 Report

Authors have very strong opinion about this manuscript. I see very few significant changes made following the first round of review. 

Reviewer 2 Report

I only have a minor correction to this highly improved manuscript.

Line 429-431 needs a reference, if that is not possible to provide, I think you should consider to erase it